# Cooling radiative forcing effect enhancement of atmospheric amines-mineral particle caused by heterogeneous uptake and oxidation

Weina Zhang[1,2], Jianhua Mai[1], Zhichao Fan[1], Yongpeng Ji[1], Yuemeng Ji[1,2], Guiying Li[1,2], Yanpeng Gao[1,2], Taicheng An[*,1,2]

[1]Guangdong-Hong Kong-Macao Joint Laboratory for Contaminants Exposure and Health, Guangdong Key Laboratory of Environmental Catalysis and Health Risk Control, Institute of Environmental Health and Pollution Control, Guangdong University of Technology, Guangzhou, 510006, China;
[2]Guangzhou Key Laboratory of Environmental Catalysis and Pollution Control, Key Laboratory of City Cluster Environmental Safety and Green Development of Ministry of Education, School of Environmental Science and Engineering, Guangdong University of Technology, Guangzhou, 510006, China

*Correspondence to*: Taicheng An (antc99@gdut.edu.cn)

**Abstract.** Warming radiative forcing effect (RFE) derived from atmospheric amines attracts lots of attentions because of their contributions to brown carbons. Herein, the enhanced influence of amines (methyl-, dimethyl-, and trimethylamine) on cooling RFE of mineral particles is first confirmed at visible wavelengths. Present results state heterogeneous uptake and oxidation reactions of atmospheric amines are feasible on mineral particle at clean/polluted conditions, which are proofed by related thermodynamics and kinetics data obtained using combined classical molecular dynamics and density function theory methods. Based on mineral particles, simple forcing efficiency (SFE) results explain that amine uptake induces at least 11.8% – 29.5% enhancement on cooling RFE of amine-mineral particles at visible wavelengths. After amines' heterogeneous oxidation, oxidized amine-mineral particles' cooling RFE are furthermore enhanced due to increased oxygen contents. Moreover, oxidized amine-mineral particles under clean condition shows 27.1% – 47.1% SFE increment at 400-600 nm, which is at least 11.3% higher than that of itself under polluted condition, due to high-oxygen-content product formation through amine autoxidation. Our results suggest cooling RFE derived from atmospheric amines can be equally important to their warming RFE on atmosphere. It is necessary to update heterogeneous oxidation mechanism and kinetics data of amines in atmospheric model in order to accurately evaluate the whole RFE caused by amines on atmosphere.

**Graphical abstract**

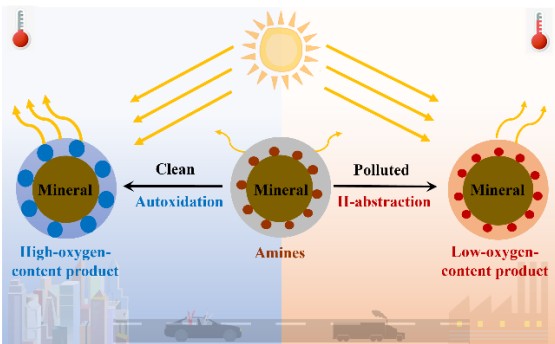

# 1 Introduction

Amines are frequently found in organic aerosols and contribute to 11.0% of $PM_{2.5}$ organic mass concentrations (Chen et al., 2021), and considered as precursors for brown carbon formations (De Haan et al., 2017; Powelson et al., 2014). Three light mass weighted amines, namely methylamine (MA), dimethylamine (DMA) and trimethylamine (TMA), are the most ubiquitous amines with the annual emission of 96.2, 38.3 and 196.0 Gg, respectively (Yu and Luo, 2014). Nucleation of these amines with sulfuric acids is verified to play a profound role in new particle formation in urban regions (Yao et al., 2018; Yin et al., 2021b; Yin et al., 2021a; Lian et al., 2020). Derived sulphate aerosols exhibits cooling radiative forced effect (RFE) due to its strong light extinction, significantly affecting the urban regional climate (Zhu et al., 2019).

Mineral particle is a dominant component of atmospheric aerosols with the annual emission of 1600 Tg (Andreae and Rosenfeld, 2008). Like sulphate aerosols, mineral particles also show strong light extinction, and thus exhibit cooling RFE on atmosphere on global scale. Moreover, anthropic organic pollutants (AOPs) including atmospheric amines are easily combined with mineral surface after uptake by mineral particles, which is confirmed by field measurements (Cheng et al., 2018), experimental observations (Huang et al., 2021) and theoretical calculations (Zhang et al., 2022). The formed AOP-mineral particle is found to exhibit distinctive RFE from corresponding AOP aerosols and mineral particles (Yu et al., 2016). For example, the RFE of black carbon-mineral particle is -112.9 $Wm^{-2}$ $\tau^{-1}$ in the low tropospheric layer, which is higher than that of the individual black carbon (-98.3 $Wm^{-2}$ $\tau^{-1}$) and mineral particle (-101.0 $Wm^{-2}$ $\tau^{-1}$) (Tian et al., 2018a). The resulting net RFE of cooling on bottom atmosphere is enhanced by 7.4% and 6.5% compared to the individual black carbon and mineral particle, respectively (Tian et al., 2018b).

AOP-mineral particles are easily oxidized through AOPs' heterogeneous oxidation reactions with atmospheric oxidants (Borduas et al., 2016; Nielsen et al., 2012; Onel et al., 2013; Onel et al., 2014). The oxidation state (OS) of AOPs thus grows higher because of oxygen addition into the parent AOPs or hydrogen removal from the initial AOPs (Kroll et al., 2015; Kroll et al., 2011). Previous studies for individual AOP aerosols confirm the light extinction/adsorption changes of oxidized aerosols/particles show strong correlations with the species and OS of AOPs. For instance, one experimental study confirms guaiacol-type aerosol's light adsorption is strengthened with the increased OS of guaiacol, but its light extinction is weakened, together leading to stronger warming RFE of oxidized aerosol compared to that of initial aerosol (Lambe et al., 2013). Another experimental study states, for α-pinene and *p*-xylene aerosols, their refractive indexes increase at low OS but decrease at high OS, finally resulting in 40% RFE enhancement of oxidized aerosol based on that of initial aerosols (He et al., 2018). Like individual AOP aerosols, the increased OS of AOP could alter the light extinction and RFE of oxidized AOP-mineral particle, which have been not attempted.

In the present study, the correlations between amine's heterogeneous oxidation reaction and RFE of amine-mineral particle are comparably explored under clean/polluted conditions. MA, DMA and TMA are employed as amine proxies considering their important contributions to SOA formation. Kaolinite (Kao) is chosen as the proxy of mineral particle because of its large emission to atmosphere (192.3 Tg $y^{-1}$) (Tang et al., 2016). First, each amine uptake by Kao particle is simulated using classical

molecular dynamics (MD) methods. Subsequently, the thermodynamics and kinetics data of the heterogeneous oxidation

reaction of each amine on Kao surface is separately calculated under polluted/clean conditions using density functional theory

(DFT). Accordingly, OS of each amine is calculated under clean/polluted conditions. Next, the light extinction of amine-Kao

particle at different OS is charactered by refractive index ($n$) and extinction coefficient ($p$) at visible wavelengths (400 – 600

nm). Finally, the RFE changes of oxidized amine-Kao particle under different conditions are estimated and compared using

simple forcing efficiency (SFE) method (Bond and Bergstrom, 2007).

## 2 Models and methods

### 2.1 Classic MD simulations

Each amine uptake by Kao particle is simulated using classical MD method. All the classical MD simulations are performed

using the package of Nanoscale MD (Phillips et al., 2005). Each amine-Kao system is equilibrated for 500 ps using *NVT*-MD

methods, where CHARMM forcefield are applied for MA, DMA and TMA (Yang et al., 2017), respectively, and Clay

forcefield is used for Kao (Tenney and Cygan, 2014). The thermostat method of Langevin is used to control temperature at

298 K. Periodic boundary condition is applied to the system. A cut-off distance of 12.0 Å is used for Lennard-Jones and real

space coulombic interactions. Particle Mesh Ewald method is also employed with the interpolation order of 6 with 1.0 Å grid

spacing.

Based on the above MD trajectories, free energy profiles of each amine-Kao particle are calculated using weighted histogram

analysis method (Kumar et al., 1995) based on umbrella sampling from MD trajectory of each amine. The details are described

in the Part 1 of **Supplement**.

### 2.2 DFT calculations

Static DFT calculations are carried out with the Vienna Ab Initio Simulation Package (Kresse and Furthmiiller, 1996; Kresse

and Furthmu ̈Ller, 1996). van-der-Waals interactions are described using the exchange-correlation functional of PBE-GGA

(Perdew et al., 1992). The electron-core interactions are described with the projector augmented wave method. Simulated

supercells are sampled with gamma-centered $3 \times 3 \times 1$ Monkhorst-Pack grids for the integration of the Brillouin Zone. To

ensure the efficiency and reliability of calculation, the kinetic cutoff energy is set at 400 eV, and the convergence criterion of

structural optimization is -0.01 eV/Å.

For DFT calculations, the initial electronic structures of amine-Kao particles are composed of Kao surface and amine molecule.

Kao surface is cleaved from Kao unit cell along (001) direction and expanded to the size of $2 \times 1$. A vacuum zone of 15 Å is

added onto Kao surface to eliminate the interaction of each layer. MA, DMA and TMA molecule are separately added above

Kao surface. Single point energies of amine-Kao ($\Delta E_{amine\text{-}Kao}$), amine ($\Delta E_{amine}$) and Kao surface ($\Delta E_{Kao}$) are separately

calculated. Therefore, the desorption energy of amine from Kao surface ($\Delta E_d$) equals to $\Delta E_{amine} + \Delta E_{Kao} - \Delta E_{amine\text{-}Kao}$. The

related results are listed in Figs. S2A-S2C.

For each amine-Kao system, potential energy surface (PES) along each plausible oxidation path is calculated. Transition states (TS) are confirmed with the method of climbing image nudged elastic band (Henkelman et al., 2000). The frequencies of reactants (RCs), TSs and products/intermediate (Pros/IMs) are calculated, respectively. Thereinto, RCs, Pros and IMs are optimized structures with no imaginary frequency, and TSs are structures with one dominant imaginary frequency. Based on PES, energy barrier ($\Delta E^{\neq}$) is $E_{TS}$ - $E_{RC}$, and reaction energy ($\Delta E_r$) is $E_{Pro/IM}$ - $E_{RC}$. Based on $\Delta E^{\neq}$ and $\Delta E_r$, $k$ is described as follows (Fernández-Ramos et al., 2007):

$$k = \sigma\kappa\frac{k_B T}{h}e^{-\frac{\Delta E^{\neq}}{RT}} , \tag{1}$$

$$k = \sigma\kappa\frac{k_B T}{h}\left(\frac{k_B T}{P_0}\right)e^{-\frac{\Delta E_r^{\neq}}{RT}} , \tag{2}$$

where equation (1) is for the first order reaction constants and equation (2) is for the second order rate constants. $\sigma$ is reaction path degeneracy, $\kappa$ is the Eckart tunneling coefficient, $k_B$ is the Boltzmann constant, $T$ is the temperature, $h$ is Planck's constant, and $P_0$ is standard atmospheric pressure. All the $\Delta E^{\neq}$ and $k$ for each reaction step are displayed in Table S1. The reaction steps, with $\Delta E^{\neq}$ lower than 20 kcal/mol, and $k$ higher than $1.38\times10^{-2}$ s$^{-1}$ (the first order reaction) or $5.68\times10^{-22}$ s$^{-1}$ molec$^{-1}$ cm$^3$ (the second order reaction), are feasible under ambient conditions (Fernández-Ramos et al., 2007).

Based on DFT results, the $n$ and $p$ of each amine-Kao particle are calculated from complex dielectric function (Fox and Bertsch, 2002) and Kramers-Kronig relation (Gajdoš et al., 2006; Wang et al., 2021).

## 2.3 Oxidation state definition

Based on the proposed heterogeneous oxidation of atmospheric amine, the oxidation degree of amine is changed because of H removal and O addition. To quantify the oxidation degree, OS of amine is determined by $n_{C-O}$ - $n_{C-H}$ - $n_{N-H}$ (Kroll et al., 2015; Kroll et al., 2011), where $n_{C-O}$, $n_{C-H}$ and $n_{N-H}$ indicate the number of C-O, C-H and N-H bonds of amine and its oxidized products, respectively. Note the OS of amine remains unchanged from RO$_2$· to RO· because no C-O, C-H or N-H bond is formed or broken. To distinguish the same OS, the suffixes of $a$ and $b$ are used. Likewise, the OS of amine remains unchanged from RO$_2$· to ROOOH. The suffixes of $c$ and $d$ are used as a distinction.

## 3 Results and discussion

### 3.1 Atmospheric amine uptake by Kao particles

Heterogeneous uptake of atmospheric amine by Kao particles at ambient condition is simulated using MD method, and the free energy profile of each amine-Kao system is calculated based on the corresponding MD trajectories (Fig. S1A). $Z$ represents the uptake distance between the mass center of single amine molecule and Kao surface. To each amine-Kao system, the free energy speeds up decreases when specific amine gets close to Kao surface, and it reaches 0 kcal/mol when amine approaches Kao surface ($Z = 3.0$ Å), indicating amine is more stable on Kao surface than that in gas phase. Such decreased trend of free

energy of amine-Kao system explains amines uptake by Kao surface is feasible at ambient condition. Meanwhile, the corresponding $\Delta E_d$ are 8.72, 6.41 and 6.85 kcal/mol for MA-Kao, DMA-Kao and TMA-Kao, respectively. Based on $\Delta E_d$, rate constants are calculated by equation (1) and equation (2). The adsorption rate constants of three amines by Kao are $6.3\times10^{-1}$, $1.3\times10^{-2}$ and $2.7\times10^{-2}$ $cm^3 \cdot molec^{-1} \cdot s^{-1}$ based on adsorption energies of -8.72, -6.41 and -6.85 kcal/mol (Fig. S5A–C). Field observations show that the concentration of Kao is $6.1\times10^{21}$ molec·$cm^{-3}$ (Scanza et al., 2015). And the concentration of amines are $2.5\times10^8$, $2.7\times10^7$ and $8.8\times10^7$ molec·$cm^{-3}$ for MA, DMA and TMA, respectively (Tan et al., 2018). Therefore, the adsorption rates of amines by Kao are in the range of $2.4\times10^7 - 1.6\times10^9$ mol·$(L\cdot s)^{-1}$, implying that these amines can be combined with Kao surface at ambient condition.

Along the uptake path, the relative concentration profile of each amine is calculated as $c(Z) = e^{-\Delta E/RT}c_{gas}$, where $\Delta E$ is read from the free energy profile (Fig. S1A), $c_{gas}$ is amine concentration in gas phase, $R$ represent ideal gas constant, and $T$ is atmospheric temperature (298 K). The calculated relative concentration profile of each amine is shown in Fig. S1B. The results show that each amine concentration keeps zero when $Z$ is larger than 4.0 Å, and the concentration sharply increases when $Z$ is smaller than 4.0 Å, indicating amine uptake by Kao surface readily occurs when amine is next to Kao particle. The peaks of each amine concentration profile are located at approximate Z = 3.0 Å, and the related peak values for MA, DMA and TMA on Kao surface are, respectively, $1.0\times10^4$, $1.0\times10^4$ and $1.8\times10^4$. These peak values are the final uptake concentration of amines by Kao surface, which are four orders higher than the related amine concentration in the gas phase. This implies heterogeneous oxidation of each amine on mineral surface can compete with their gaseous oxidations, especially in the regions with abundant mineral particles (Uno et al., 2011).

To find out why each amine is more stable on Kao surface than that in gas phase, the interactions between amine and Kao surface are analyzed using charge density difference (CDD) method. As shown in Figs. S2A – S2C, the regions with increased electron density are shown in yellow, and the regions with decreased electron density are shown in blue. In each amine-Kao particle, N atom of relevant amine exhibits increased trends of electron density, and the regions around H atom of Kao surface presents decreased trends of electron density. This means strong combination exists between N of each amine and H of Kao surface. Moreover, the measured distances between N and H are 1.81 Å in MA-Kao particle, 1.75 Å in DMA- and TMA-Kao particles (Figs. S2A – S2C), which are all shorter than 2.5 Å. By combing CCD and measured distance results, hydrogen bond between N and H atoms in amine-Kao particle is confirmed, contributing to stability of each amine on Kao surface. This is also observed in other AOP-mineral particles (Zeitler et al., 2017; Romanias et al., 2016). During the following oxidation reactions, the above hydrogen bonds keep amine or its derived products combined with Kao particle, offering the possibility for amine to proceed heterogeneous oxidation on Kao surface, affecting the extinction and RFE of amine-Kao particle.

**3.2 Heterogeneous oxidation mechanism of atmospheric amines**

On Kao surface, the heterogeneous oxidation reactions of atmospheric amines are initiated by ·OH. ·OH first binds with amines to form pre-reactive complex, and the rate constants are $1.0\times10^7$, $1.6\times10^7$ and $1.7\times10^8$ $cm^3 \cdot molec^{-1} \cdot s^{-1}$ based on reaction energies of -18.56, -18.82 and -20.23 kcal/mol (Fig. 1A).The concentration of ·OH is $2.4\times10^6$ molec·$cm^{-3}$ in the atmosphere

(Tan et al., 2018). Therefore, based kinetics calculation, the formation rate of the pre-reactive complex is in the range of 10.3 – 59.7 mol·(L·s)$^{-1}$. For MA and DMA, they each have two possible initial oxidation routes, i.e., H-abstraction routes from their methyl and amido groups. TMA only has the former H-abstraction route from methyl group because its hydrogen in amido group is completely substituted by methyl group. The relevant PESs of two routes are compared in Fig. 1. All energy barriers and rate constants are summarized in Table S1. We first pay attention on MA's initial heterogenous oxidation. Based on PES, the related $k$ for methyl route is calculated as $1.27 \times 10^5$ s$^{-1}$ molec$^{-1}$ cm$^{-3}$, which is 6 orders higher than that of amido route ($1.24 \times 10^{-1}$ s$^{-1}$ molec$^{-1}$ cm$^{-3}$). Accordingly, methyl route is dominant for MA's initial heterogenous oxidation reaction. Agreements are observed in DMA and TMA's heterogeneous oxidation reactions. However, for amines in the gas phase, the amido route is dominant (Onel et al., 2013). This is because amido site is occupied by H atom of Kao surface (Figs. S2A – S2C). Accordingly, Kao surface is found to alter the initial oxidation route of amines. Moreover, the present rate constants of heterogenous initial oxidation reactions are much larger than those of the corresponding gaseous ones (Table S2). For instance, the related $k$ for heterogenous initial oxidation reaction of DMA is calculated as 9.22 s$^{-1}$ molec$^{-1}$ cm$^3$, which is 11 orders higher than that of the gaseous one (Onel et al., 2013). Therefore, our results declare Kao surface significantly accelerates the initial oxidation reactions of atmospheric amines at ambient condition.

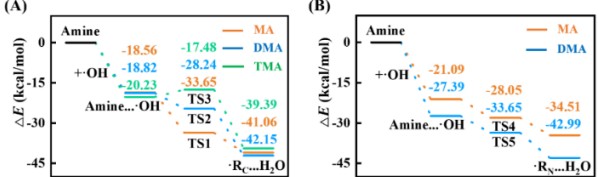

**Figure 1** Initial oxidation reactions of amines with ·OH. PESs of H-abstraction routes (A) from methyl group and (B) from amido group of MA, DMA and TMA, respectively.

Subsequently, ·R$_C$ of each amine is separately converted into the corresponding amine-RO$_2$· through O$_2$-addition, which is barrierless reaction according to PESs (Fig. S3). Amine-RO$_2$· is further oxidized by different atmospheric oxidants under polluted and clean conditions. Under polluted condition, each amine-RO$_2$· reacts with NO/O$_2$ following the steps of NO-addition and H-abstraction in order. Based on corresponding PESs (Fig. 2), the NO-addition step is barrierless and the energy barrier of H-abstractions step is positive, which are both feasible at ambient condition. Low-oxygen-content product, i.e., RNCHO, is finally produced from each amine under polluted condition. Compared to initial amines, the OS of each RNCHO increases by 4 under polluted condition, resulted from O-addition to amine and H-removal from amine.

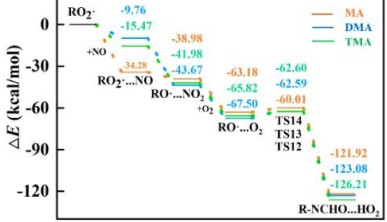

**Figure 2** PESs of subsequent oxidation reactions of amines' RO$_2$· with NO/O$_2$ under polluted condition.

Under clean condition, MA- or DMA-RO$_2\cdot$ could be further oxidized by O$_2$ through autoxidation or H-abstraction routes. First, the feasibility of MA-RO$_2\cdot$'s oxidation by O$_2$ is investigated. Based on PES of MA-RO$_2\cdot$'s autoxidation route (orange line in Fig. 3A), the first H-transfer step is found to be the rate-limiting step with high $\Delta E^{\neq}$ of 27.38 kcal/mol, and the derived $k17$ is obtained as $1.34\times10^{-7}$ s$^{-1}$, indicating the autoxidation route is not feasible at ambient condition. Similarly, H-abstraction route of MA-RO$_2$ is not feasible due to its high $\Delta E^{\neq}$, either (Fig. S4). Like MA-RO$_2\cdot$, the second H-transfer step of DMA-RO$_2\cdot$'s autoxidation route (blue line in Fig. 3A) and H-abstraction route of DMA-RO$_2\cdot$ (Fig. S4) is not feasible due to its high $\Delta E^{\neq}$.

But the MA- or DMA-RO$_2\cdot$ as an intermediate in the autoxidation route can be oxidized by $\cdot$OH to produce high-oxygen-content product (Berndt et al., 2022), i.e., NH$_2$CH$_2$OOOH or HOOCH$_2$NHCH$_2$OOOH (Fig. 3B). This is a barrierless reaction and $\Delta E_r$ is -27.16 or -32.06 kcal/mol, indicating the $\cdot$OH-addition of MA- or DMA-RO$_2\cdot$ is feasible at ambient condition. For MA and DMA, R-NCH$_2$OOOH with high-oxygen-content is the product under clean condition. Compared to initial MA/DMA, OS of oxidized MA and DMA increases by 2 and 4 (Fig. 4).

Different from MA- or DMA-RO$_2\cdot$, TMA-RO$_2\cdot$ can only proceed autoxidation under clean condition. This is because TMA's hydrogens are completely substituted by methyl groups, leading to no more H-abstraction sites on nitrogen of TMA. TMA-RO$_2\cdot$'s autoxidation is composed of H-transfer steps and O$_2$-addition steps. Based on PES (green line in Fig. 3A), the highest $\Delta E^{\neq}$ of three H-transfer steps is obtained as 14.15 kcal/mol, and derived $k$ is $2.23\times10^3$ s$^{-1}$, indicating the autoxidation is feasible at ambient condition. The high-oxygen-content product, i.e., (HOOCH$_2$)$_2$NCHO, is finally produced from TMA under clean condition. Compared to initial TMA, OS of oxidized TMA increases by 8 (Fig. 4), due to 5 O atom additions to TMA from 3 O$_2$-addition steps.

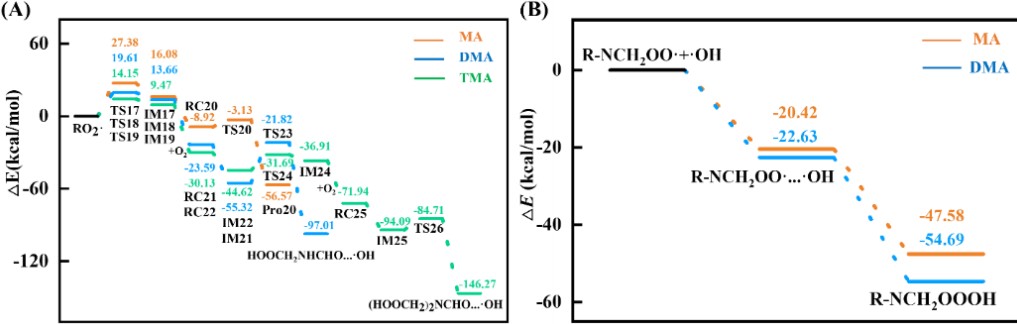

**Figure 3** PESs of subsequent (A) autoxidation and (B) $\cdot$OH-addition of amines' RO$_2\cdot$ under clean condition.

Note that NO concentration is important to determine amines heterogeneous oxidation mechanisms. As proposed under polluted conditions, amine-RO$_2\cdot$ and NO form a pre-reaction complex RO$_2\cdot$…NO before H-abstraction reaction (Fig.2). The corresponding formation rate $r_{RO2\cdot…NO} = k_{RO2\cdot…NO}$ [RO$_2\cdot$][NO]. Under clean conditions, the amine-RO$_2\cdot$ carries on H-transfer and forms $\cdot$ROOH (Fig.3A). The corresponding formation rate of $\cdot$ROOH is $r_{\cdot ROOH} = k_{\cdot ROOH}$[RO$_2\cdot$]. $r_{RO2\cdot…NO}/r_{\cdot ROOH} > 10^3$ indicates H-abstraction reaction is dominant under polluted condition, where NO concentration should be higher than $k_{\cdot ROOH}/k_{RO2\cdot…NO}\times10^3$. Conversely, $r_{RO2\cdot…NO}/r_{\cdot ROOH} < 10^{-3}$ implies autoxidation reaction is dominant under clean conditions, where NO concentration is lower than $k_{\cdot ROOH}/k_{RO2\cdot…NO}\times10^{-3}$. Specifically, $k_{RO2\cdot…NO} = 3.5\times10^6$ cm$^3\cdot$molec$^{-1}\cdot$s$^{-1}$ and $k_{\cdot ROOH} =$

$5.2\times10^{-3}$ s$^{-1}$ for MA, $k_{RO2\cdots NO} = 3.7$ cm$^3\cdot$molec$^{-1}\cdot$s$^{-1}$ and $k_{-ROOH} = 2.6\times10^{-2}$ s$^{-1}$ for DMA, as $k_{RO2\cdots NO} = 5.6\times10^4$ cm$^3\cdot$molec$^{-1}\cdot$s$^{-1}$ and $k_{-ROOH} = 2.6\times10^2$ s$^{-1}$ for TMA system. Therefore, NO concentration is determined as higher than $1.5\times10^{-6}$, 7.2 and 4.7 molec$\cdot$cm$^{-3}$ under polluted condition and below $1.5\times10^{-12}$, $7.2\times10^{-6}$ and $4.7\times10^{-6}$ molec$\cdot$cm$^{-3}$ under clean conditions, respectively. Therefore, for a mixture source of the three amines, all amines proceed H-abstraction reactions when NO concentration is higher than 7.2 molec$\cdot$cm$^{-3}$ and carry on autoxidation reaction when NO concentration is smaller than $1.5\times10^{-12}$ molec$\cdot$cm$^{-3}$. Field observation shows that the concentration of NO in polluted conditions ranges from $4.1\times10^6 - 1.9\times10^{13}$ molec$\cdot$cm$^{-3}$. And when the concentration of NO is in the range of $5.2\times10^{-9} - 7.8\times10^{-3}$ molec$\cdot$cm$^{-3}$, it is considered as a relatively clean condition (Tan et al., 2018). The corresponding field measurement results are consistent with the theoretical calculation range.

In addition, Kao particle is also found to reduce $\Delta E^{\neq}$ and increase $k$ of heterogeneous autoxidation of TMA referred to gaseous autoxidation (Table. S2). Specifically, $\Delta E^{\neq}$ of three H-transfer steps during heterogeneous autoxidation of TMA are separately reduced by 1.85 – 3.65, 5.27 – 5.29 and 8.72 – 10.72 kcal/mol compared to different reported results, and the corresponding $k$ are obtained as 3, 4 and 8 orders higher than those in gas phase (Ma et al., 2021; Moller et al., 2020), respectively. Therefore, compared to gaseous autoxidation of TMA, heterogeneous autoxidation is more feasible at ambient condition, leading to abundant high OS products in oxidized amine-Kao particles. Moreover, during the whole heterogeneous oxidation under clean/polluted conditions, all oxidant intermediates and products keep combined with Kao surface (Fig. S5), leading to continuous influence on optical properties and RFE of amine-Kao particles.

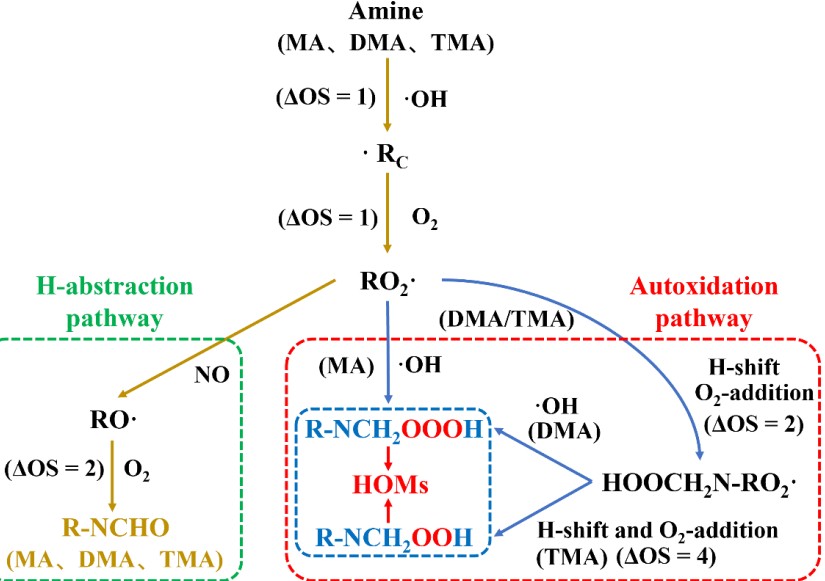

**Figure 4** Proposed aging mechanism and related *OS* value changes for amine-Kao mixed particles under polluted (brown line) and clean (blue line) conditions.

### 3.3 Refractive index (*n*) and extinction coefficient (*p*)

Based on the proposed heterogeneous oxidation mechanisms (Fig. 4), the correlation of *n*/*p* of each oxidized amine-Kao particles and OS is separately calculated at visible wavelengths (400–600 nm). First, *p* changes of each amine-Kao particles are compared under clean/polluted conditions in Fig. 5. *p* of each amine-Kao particles keeps equal to that of Kao particle, implying amine uptake has no effect on Kao particle's light transmission. For example, at 400 nm, *p* is equal to 0.0012 for MA-, DMA- and TMA-Kao (Figs. 5A–5C), respectively, which is equal to that of Kao particle. As amine oxidation starts, *p* of each oxidized amine-Kao particle slightly increases. For instance, at 400 nm, *p* of oxidized TMA-Kao particle under polluted condition increases to 0.002 at OS = -1 from 0.001 at OS = -9 (Fig. 5C), with a small $\Delta p = 0.001$ that can be ignored. One exception that *p* increases to 0.012 at OS = -8 exists, due to $\cdot R_C$ formation. However, $\cdot R_C$ is rapidly converted to molecule products, resulting to a temporary increase of *p*. *p* of oxidized MA- and DMA-Kao also present similar changes under clean/polluted conditions (Figs. 5A-5B). Accordingly, the heterogeneous uptake and oxidation of each amine have no influence on amine-Kao particle's light transmission at ambient condition.

Next, attentions are paid on *n* changes of each amine-Kao particle. Like *p* changes with wavelengths, *n* of each oxidized amine-Kao particle decreases with the increased wavelength (Fig. S6A-S6C), which is similar to that of that observed for individual AOP aerosols (Flores et al., 2014; He et al., 2022; Jiang et al., 2019). However, different from *p*, *n* of each amine-Kao particle is higher than that of Kao particle (Figs. S6A – S6C), implying amine uptake enhances Kao particle's light extinction.

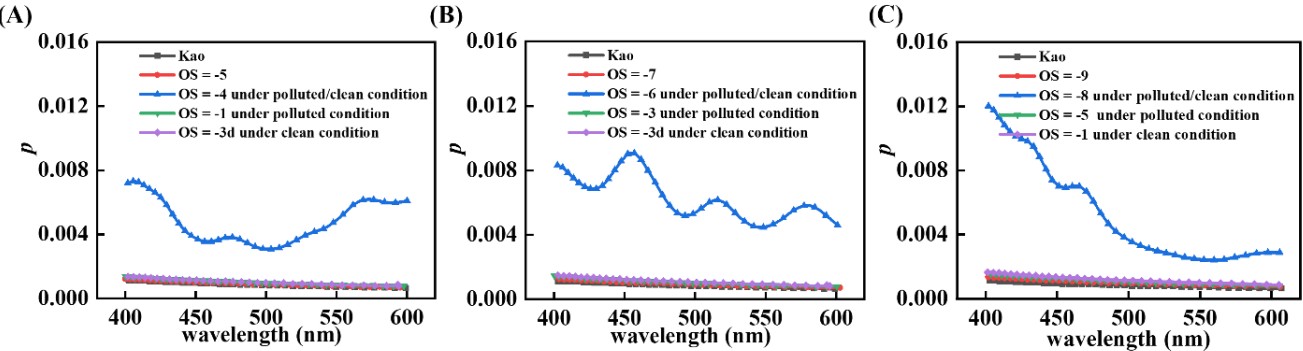

**Figure 5** *p* changes of each amine-Kao particle with increased wavelength (400 – 600 nm) under polluted and clean conditions. Results for (A) MA-Kao, (B) DMA-Kao and (C) TMA-Kao particles, respectively.

Moreover, *n* of TMA-Kao > *n* of DMA-Kao > *n* of MA-Kao, which may be determined by molecular mass of amines. As heterogeneous oxidation proceeds, *n* of each oxidized amine-Kao particle further increases with increased OS (Fig. 6), and shows distinctive profiles caused by different oxidation mechanisms. We first concentrate on *n* changes caused by H-abstraction. Present results have declared that MA-Kao, DMA-Kao and TMA-Kao particle under polluted conditions are oxidized through H-abstraction reactions. Figs. 6A-6C display H-abstraction induces very small *n* increments of these oxidized amine-Kao particle. For instance, for oxidized TMA-Kao particle, $\Delta n$ at OS = -5 is 0.003 at 589 nm (Fig. 6C). Similar agreements are observed in oxidized MA- and DMA-Kao particles. By contrast, autoxidation causes large *n* increments of

oxidized amine-Kao particle. As shown in Fig. 6F, TMA-Kao's $\Delta n$ at OS = -1 is 0.017 at 589 nm, which is at least 5 times larger under clean condition than that of oxidized TMA-Kao particle under polluted condition. Like TMA-Kao, MA- and DMA-Kao particles also have a large $\Delta n$ under clean condition. The $\Delta n$ of MA- and DMA-Kao particles at OS = -3d is 0.016 and 0.017 at 589 nm under clean condition, which is 0.012 and 0.016 higher than under polluted condition. Large $\Delta n$ of oxidized amine-Kao particle is caused by high-oxygen-content product with high level of OS, resulting in strong extinction of amine-Kao particle at clean condition.

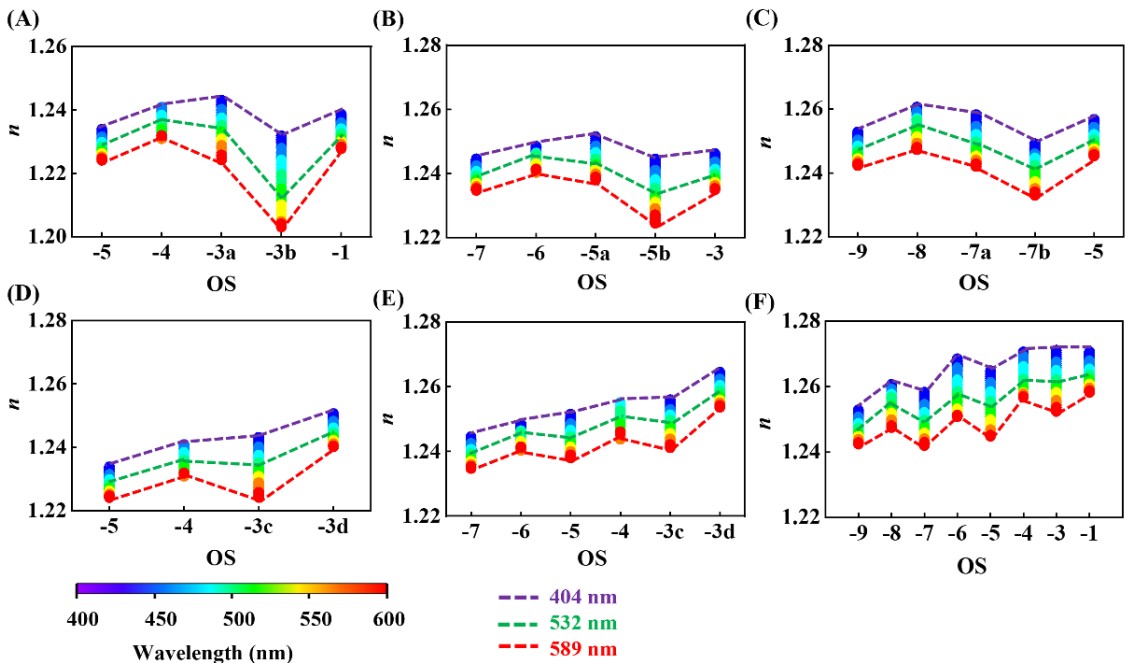

**Figure 6** The relationship between $n$ and $OS$ in the wavelength range of 400-600 nm under polluted (upper panel) and clean (under panel) conditions. Related profiles are for (A, D) MA-Kao (B, E) DMA-Kao and (C, F) TMA-Kao mixed particles, respectively. Three trendlines in colors of purple, green and red are marked at wavelengths of 404, 532 and 589 nm, respectively.

### 3.4 Quantitative structure–property relationship analysis

To explain why $n$ of oxidized amine-Kao particles increases more under clean condition than that of any amine-Kao particles under polluted condition, the molecular structure parameter changes with the increased OS under different conditions are compared using quantitative structure–property relationship (QSPR) method reported from the reference (Redmond and Thompson, 2011). The molecular structure parameters include unsaturation ($\mu$), polarizability ($\alpha$) and molar mass ($M$) of all oxidation products with different OS, which are all positively associated with $n$ of amine-Kao particles. Thereinto, $\alpha = 1.51(\#C) + 0.17(\#H) + 0.57(\#O) + 1.05(\#N) + 0.32$, and $\mu = (\#C + 1) - 0.5(\#H - \#N)$, where #atom represents the number of specific atoms. The increments of these parameter are written as $\Delta\mu$, $\Delta\alpha$ and $\Delta M$, respectively.

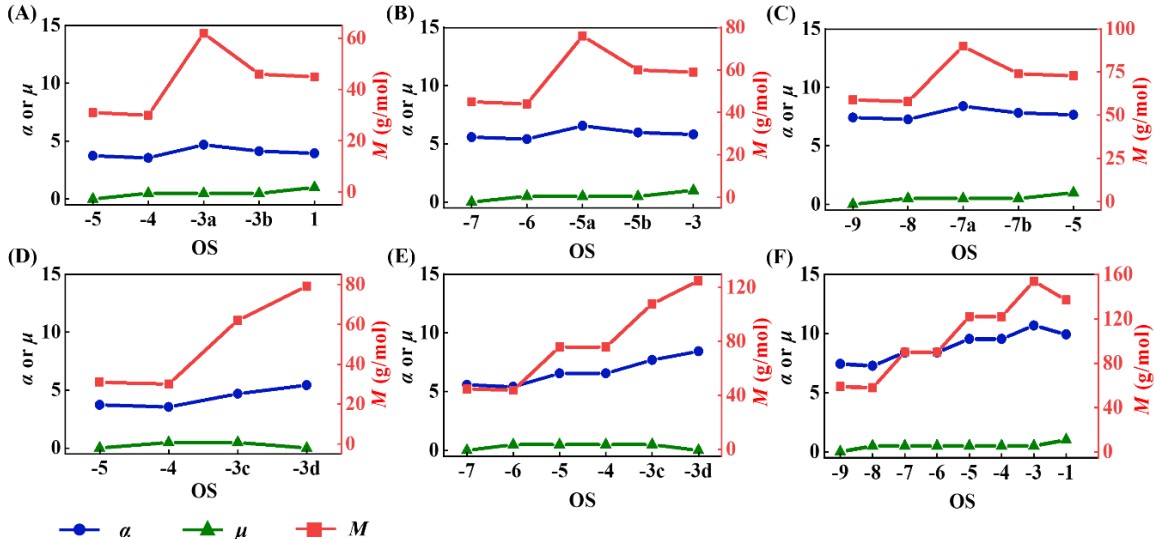

**Figure 7** The molecular structure parameter changes with the increased OS during (A-C) H-abstraction and (D-F) autoxidation.

Figs. 7A–7C illustrate the structure parameter changes of MA, DMA and TMA under polluted conditions. Specifically, as $\Delta OS = 4$, $\Delta\alpha$, $\Delta M$ and $\Delta\mu$ of each amine are obtained as 0.23, 14 g/mol and 1, respectively. These small increments of molecular mass, polarizability and unsaturation are accounted for slight enhancement of $n$ of oxidized amine-Kao particles, which are derived from formations of RNCHO with low oxygen content under polluted condition. Moreover, due to the same H-removal and O-addition levels of RNCHO based on related amine, these parameter increments are equal. This suggests, under polluted condition, $n$ increments of oxidized amine-Kao particles are more determined by oxidation process but little related to amine structures.

More attentions are paid on the structure parameter changes of amine under clean condition. Compared to initial TMA, $\Delta\alpha$ and $\Delta M$ are obtained as 2.51 and 78 g/mol separately (Fig. 7F), which are about 11 and 6 times larger those under polluted condition. Similarly, MA or DMA's $\Delta\alpha$ and $\Delta M$ under clean condition are large than polluted condition. Amine under clean condition is confirmed to carry on autoxidation on Kao surface in this work, producing high-oxygen-content product ($NH_2CH_2OOOH$, $HOOCH_2NHCH_2OOOH$ and $(HOOCH_2)_2NCHO$). Therefore, significant increments of molecule mass and polarizability together contribute to large $n$ of oxidized amine-Kao particle, resulting in strong extinction under clean condition. This is also observed in oxidized naphthalene aerosol (He et al., 2022).

### 3.5 Enhanced simple forcing efficiency

Based on obtained $n$ increments (Fig. 6), SFE (Bond and Bergstrom, 2007) is calculated to quantify the influence of heterogeneous uptake and oxidations of each amine on Kao particle under clean/polluted conditions. $\Delta$SFE is defined as the ratio of SFE of oxidized amine-Kao particles to that of individual Kao particle. Note, at the same wavelength and OS, $p$ is at least 3 orders smaller than $n$ of the same amine-Kao particle. For example, for TMA-Kao particle at 600 nm, the $p$ is less than

0.001 at any OS of TMA (Fig. 5C), and the corresponding $n$ keeps larger than 1.25 (Figs. 6C and 6F). Therefore, $p$ is ignored when evaluating SFE of amine-Kao particles. The related derivations of SFE and ΔSFE are given in Part 2 in SI.

Amine uptake enhances cooling RFE of amine-Kao particle at 400 – 600 nm, which is proofed by high levels of ΔSFE profiles for each amine-Kao particle (black curves in Fig. S7). Specifically, ΔSFE is obtained as 12.0% for MA-Kao particle, 22.0% for DMA-Kao particle and 29.5% for TMA-Kao particle, respectively. ΔSFE is furthermore increased after amine oxidation under polluted/clean conditions. For example, at 400 – 600 nm, ΔSFE of oxidized TMA-Kao particle is obtained as 32.0% – 33.6% under polluted condition (red curve in Fig. S7C) and 45.6% – 47.1% under clean condition (blue curve in Fig. S7C). The same increased also occurs in MA- or DMA-Kao oxidation. ΔSFE of oxidized MA- or DMA-Kao particle is obtained as 15.4% – 16.2% or 22.1% - 23.1% under polluted condition (red curve in Fig. S7A-B) and 27.1% – 27.5% or 40.9% - 41.4% under clean condition (blue curve in Fig. S7A-B). Compared to that under polluted condition, larger ΔSFE of oxidized amine-Kao particle under clean condition is attributed to stronger particle's extinction, due to high-oxygen-content product formation. In summary, heterogeneous uptake and oxidation of amines on Kao surface are completely investigated under clean/polluted conditions using DFT methods, and their enhanced effects on cooling RFE of amine-Kao particles on atmosphere are first confirmed. Like other AOPs (Borduas et al., 2016; Nielsen et al., 2012; Onel et al., 2013; Onel et al., 2014), amine uptake by Kao particle is feasible, and heterogeneous oxidation reactions are found to be more competitive than gaseous oxidation reactions, leading to an increase in oxygen content of amine-Kao particles in atmosphere. Afterwards amine uptake, compared to Kao particles, SFE is enhanced by 12% – 29.5% for amine-Kao particles, suggesting amine uptake enhances cooling RFE of Kao particle on atmosphere. Amine's heterogeneous oxidation reactions are found to furthermore enhance Kao's cooling RFE, due to the increase in oxygen content of amine-Kao particles. More importantly, under clean condition with low NO concentration, SFE of oxidized amine-Kao particle is enhanced by 27.1% – 47.1%, which is at least 11.3% – 18.8% higher than that of amine-Kao particle under polluted condition. The significant cooling RFE enhancement of oxidized amine-Kao particles is resulted by high-oxygen-content product generated from amine autoxidation. This conclusion can be deduced to other AOPs with complete hydrogen substitution by methyl groups, such as triethylamine (Ma et al., 2021), dimethyl sulfide (Berndt et al., 2019; Wu et al., 2015; Veres et al., 2020) and dimethyl ether(Wang and Wang, 2016), etc, which prefer to carry on autoxidation under clean condition.

Amines are commonly confirmed as precursors for brown carbon formations (De Haan et al., 2017; Powelson et al., 2014), and lots of attentions are paid on their contributions to light-adsorbing and warming RFE on atmosphere. However, our results propose amines can enhance cooling RFE on atmosphere, especially under clean conditions, through heterogeneous uptake and oxidation reactions on mineral particles. Considering large amounts of mineral particles in atmosphere (Andreae and Rosenfeld, 2008) and their inherent cooling RFE on global scale, the proposed cooling RFE derived from atmospheric amines can be equally important to their warming RFE on atmosphere. Therefore, it is necessary to update heterogeneous oxidation mechanism and kinetics data of amines in order to accurately evaluate related comprehensive RFE on atmosphere.

## 4 Conclusions

In summary, heterogeneous uptake and oxidation of amines on Kao surface are completely investigated under clean/polluted conditions using DFT methods, and their enhanced effects on cooling RFE of amine-Kao particles on atmosphere are first confirmed. Like other AOPs (Borduas et al., 2016; Nielsen et al., 2012; Onel et al., 2013; Onel et al., 2014), amine uptake by Kao particle is feasible, and heterogeneous oxidation reactions are found to be more competitive than gaseous oxidation reactions, leading to an increase in oxygen content of amine-Kao particles in atmosphere. Afterwards amine uptake, compared to Kao particle, SFE is enhanced by 12% – 29.5% for amine-Kao particles, suggesting amine uptake enhances cooling RFE of Kao particle on atmosphere. Amine's heterogeneous oxidation reactions are found to furthermore enhance Kao's cooling RFE, due to the increase in oxygen content of amine-Kao particle. More importantly, under clean condition with low NO concentration, SFE of oxidized amine-Kao particles is enhanced by 27.1% – 47.1%, which is at least 11.3% – 18.8% higher than that of amine-Kao particle under polluted condition. The significant cooling RFE enhancement of oxidized amine-Kao particles is resulted by high-oxygen-content product generated from amine autoxidation. This conclusion can be deduced to other AOPs with complete hydrogen substitution by methyl groups, such as triethylamine (Ma et al., 2021), dimethyl sulfide (Berndt et al., 2019; Wu et al., 2015; Veres et al., 2020) and dimethyl ether (Wang and Wang, 2016), etc, which prefer to carry on autoxidation under clean condition.

Amines are commonly confirmed as precursors for brown carbon formations (De Haan et al., 2017; Powelson et al., 2014), and lots of attentions are paid on their contributions to light-adsorbing and warming RFE on atmosphere. However, our results propose amines can enhance cooling RFE on atmosphere, especially under clean conditions, through heterogeneous uptake and oxidation reactions on mineral particles. Considering large amounts of mineral particles in atmosphere (Andreae and Rosenfeld, 2008) and their inherent cooling RFE on global scale, the proposed cooling RFE derived from atmospheric amines can be equally important to their warming RFE on atmosphere. Therefore, it is necessary to update heterogeneous oxidation mechanism and kinetics data of amines in order to accurately evaluate related comprehensive RFE on atmosphere.

**Author contributions.** WZ and TA designed the research goals, aims and methodology. JM and ZF contributed to calculations and data processing. WZ analysed the data and wrote the manuscript. WZ, JM, ZF, YJ, YMJ, YG, GL and TA proofread and commented on the paper.

**Supplement.** The supplement related to this article is available online at:

**Competing interests.** The authors declare that they have no conflict of interest.

**Disclaimer.** Publisher's note: Copernicus Publications remains neutral with regard to jurisdictional claims made in the text, published maps, institutional affiliations, or any other geographical representation in this paper. While Copernicus Publications makes every effort to include appropriate place names, the final responsibility lies with the authors.

**Financial support.** This work is supported by NSFC (42020104001, 42277081 and 42077189), the Basic and Applied Basic Research Fund Project of Guangdong Province (2024A1515012691 and 2019B151502064), the National Key Research and
355 Development Program of China (2022YFC3105600), and Guangdong Provincial Key R&D Program (2022-GDUT-A0007).

**Review statement.**

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
