# Peer review of "Cooling radiative forcing effect enhancement of atmospheric aminesmineral particle caused by heterogeneous uptake and oxidation"

_EGUsphere, 2024_

## Author Response (AR1)

**Manuscript ID: egusphere-2024-1048**

**Title:** Cooling radiative forcing effect enhancement of atmospheric amines-mineral particle caused by heterogeneous uptake and oxidation

**The corresponding authors:** Prof. Taicheng An

**Dear Anonymous Referee #1,**

Thank you for the helpful and valuable review and comment. We have made careful revisions on the original manuscript according to your kind and helpful comments. The changed sentences have been marked as red color in the revised manuscript. Below is our point-by-point response to your comments:

**General comment:** This manuscript reports the theoretical cooling radiative forced effect (RFE) derived from heterogeneous uptake and oxidation of atmospheric amines (i.e., methylamine, dimethylamine, and trimethylamine) on Kaolinite particles under clean and polluted conditions. Overall, the results show amines can enhance cooling RFE via heterogeneous uptake and oxidation of amines on mineral particles, especially under clean conditions (i.e., low $NO_x$ level). Furthermore, increased oxygen contents of oxidation products of amines on the surface of Kaolinite particles can also reinforce the cooling RFE. The authors also suggest cooling RFE can be comparable with their warming RFE in the atmosphere.

**Response:** We appreciate reviewer's suggestive comments. Each comment is carefully address and all the modifications are marked in red in the revised manuscript.

**Question** 1. Atmospheric heterogeneous oxidation of amines is crucial in term of the formation of brown carbon and the optical properties of aerosols. However, the conclusions of this manuscript were based on the theoretical calculations. More atmospherically relevant evidence should be added to verify these results.

**Response:** we agree with you that atmospherically relevant evidence is very important to verify these results. Therefore, in the revised manuscript, we supplied the adsorption rate constants of three amines by Kao particle (Line 122-127), rate of initial OH-initiated reaction (Line 150-154) and NO concentration under clean/polluted conditions (Line 199-214) based on field measurement data of atmospherically

relevant levels of amines, Kao particles, OH and NO, respectively. these newly added calculations furthermore verify that our proposed heterogeneous oxidation mechanisms of amines on Kao particles is feasible in atmosphere.

As for RFE sections, we did not find experimental or observed studies on the effect of amine oxidation on particles' RFE. However, an experiment study found that the optical scattering properties of aerosols are enhanced with the increase of oxygen content in the oxidation component (Redmond and Thompson, 2011). In our work, amine-Kao mixed particle is also confirmed to increase oxygen content under clean/polluted conditions. Therefore, we used the same method to evaluate RFE (Redmond and Thompson, 2011) of oxidized amine-Kao mixed particles here.

**Question** 2. In the main text and SI, many details of theoretical calculations were presented. However, the concentrations/levels of amines, OH radicals, NO, Kao particles should also be listed.

**Response:** The concentrations/levels of amines and Kao particles are $2.5\times10^8$, $2.7\times10^7$, $8.8\times10^7$ and $6.1\times10^{21}$ molec·cm$^{-3}$, which are observed results from Tan et al., 2018 and Scanza et al., 2015, respectively. They are supplied in Line 124-125. Moreover, we calculated the adsorption rate constants of three amines by Kao, which locates in $6.3\times10^{-1}$, $1.3\times10^{-2}$ and $2.7\times10^{-2}$ cm$^3$·molec$^{-1}$·s$^{-1}$ (Line 122-123). The above results together confirm that the adsorption rate of amines and Kao range in $2.4\times10^7$ – $1.6\times10^9$ mol·(L·s)$^{-1}$, implying that these amines can be combined with Kao surface at ambient condition (Line 125-127).

The concentration of ·OH is $2.4\times10^6$ molec·cm$^{-3}$ in the atmosphere (Tan et al., 2018). According to this, the formation rate of the first step of OH-initiated oxidation is in the range of 10.3 – 59.7 mol·(L·s)$^{-1}$. The corresponding descriptions are provided in Line 150-154.

NO concentration is smaller than $1.5\times10^{-12}$ molec·cm$^{-3}$ under clean condition and higher than 7.2 molec·cm$^{-3}$ under polluted condition. The two NO concentrations are determined by our thermodynamics and kinetics data (Line 209-211), which are consistent with atmospheric NO concentrations obtained

from Tan et al., 2018. The detailed discussions of NO concentration calculations are provided in Line 199-214.

**Question** 3. Line 28-29: it was stated that "Amines are frequently found in organic aerosols and contribute to 20% of $PM_5$ mass concentrations (Huang et al., 2022; Silva et al., 2008)". Please carefully check these two cited papers, it may be exaggerated that the contribution of amines to $PM_{2.5}$ mass loadings can be up to 20%.

**Response:** The reference (Chen et al., 2021) found that the organic mass concentration of amines in $PM_{2.5}$ is 11.0%. We have corrected in line 28.

**Question** 4. Line 114 and 163, the descriptions of "ambient condition" and "polluted and clean conditions" are vague. Please fill them in detail.

**Response:** Ambient condition indicates temperature = 298 K and pressure = 1 bar. NO concentration is smaller than $1.5 \times 10^{-12}$ molec·$cm^{-3}$ under clean condition and higher than 7.2 molec·$cm^{-3}$ under polluted condition. The two NO concentrations are determined by our thermodynamics and kinetics data (Line 209-211), which are consistent with atmospheric NO concentrations obtained from Tan et al., 2018 . The detailed discussions of NO concentration calculations are provided in Line 199-214.

**Reference**

Bond, T. C. and Bergstrom, R. W.: Light Absorption by Carbonaceous Particles: An Investigative Review, Aerosol Science and Technology, 40, 27-67, 10.1080/02786820500421521, 2007.

Redmond, H. and Thompson, J. E.: Evaluation of a quantitative structure-property relationship (QSPR) for predicting mid-visible refractive index of secondary organic aerosol (SOA), Physical Chemistry Chemical Physics, 13, 6872-6882, 10.1039/c0cp02270e, 2011.

Scanza, R. A., Mahowald, N., Ghan, S., Zender, C. S., Kok, J. F., Liu, X., Zhang, Y., and Albani, S.: Modeling dust as component minerals in the Community Atmosphere Model: development of framework and impact on radiative forcing, Atmospheric Chemistry and Physics, 15, 537-561, 10.5194/acp-15-537-2015, 2015.

Tan, Z., Rohrer, F., Lu, K., Ma, X., Bohn, B., Broch, S., Dong, H., Fuchs, H., Gkatzelis, G. I., Hofzumahaus, A., Holland, F., Li, X., Liu, Y., Liu, Y., Novelli, A., Shao, M., Wang, H., Wu, Y., Zeng, L., Hu, M., Kiendler-Scharr, A., Wahner, A., and Zhang, Y.: Wintertime photochemistry in Beijing: observations of ROx radical concentrations in the North China Plain during the BEST-ONE campaign, Atmospheric Chemistry and Physics, 18, 12391-12411, 10.5194/acp-18-12391-2018, 2018.

**Dear Anonymous Referee #1,**

Thank you for the helpful and valuable review and comment. We have made careful revisions on the original manuscript according to your kind and helpful comments. The changed sentences have been marked as red color in the revised manuscript. Below is our point-by-point response to your comments:

**General comment:** The authors systematically studied the influence of heterogeneous oxidation reaction of amines on RFE of mixing amine-mineral particles at visible wavelengths using density functional theory calculations. They reported more significant enhancement of RFE caused by heterogeneous autoxidation of trimethylamine. Their results highlight that cooling RFE derived from atmospheric amines can be equally important to their warming RFE on atmosphere. The outcomes are very helpful to provide the mechanistic and kinetic data of amines in the atmospheric models for the accurate evaluation of RFE. This paper is well organized and clearly written, and the topic is very interesting and appropriate for **Atmospheric Chemistry and Physics**. Several comments on this work need to be addressed to improve the quality of the manuscript before publication:

**Response:** We appreciate reviewer's suggestive comments. Each comment is carefully addressed and all the modifications are marked in red in the revised manuscript.

**Question** 1. Grammar mistakes:

- There are two predicates for "…simple forcing efficiency (SFE) results explain amine uptake induces at least 11.8% – 29.5%..." in lines 17-18.

- In line 28, a space is missing between "formations" and "(" in the sentence "… considered as precursors for brown carbon formations(De Haan et al., 2017; Powelson et al., 2014)".

- In line 42, another space is missing between "Wm$^{-2}$ $\tau^{-1}$)" and "(Tian" in the sentence "… and mineral particle (-101.0 Wm$^{-2}$ $\tau^{-1}$)(Tian et al., 2018a)".

- In line 75, "is" should be corrected into "are".

- In line 120, "desorption energies ($\Delta E_d$)" is repeated, which has been described in MEHTODS.

- In line 139, "this" should be corrected into "This".
- In line 149, "which" should be "which is".

**Response:** We have corrected these mistakes in the preprint version.

(1) In line 17, the sentence "…simple forcing efficiency (SFE) results explain amine uptake induces at least 11.8% – 29.5%..." have modified to "simple forcing efficiency (SFE) results explain that amine uptake induces at least 11.8% – 29.5%".

(2) In line 29, a space has added between "formations" and "(De Haan et al., 2017; Powelson et al., 2014)".

(3) In line 42, a space has added between "Wm$^{-2}$ $\tau^{-1}$)" and "(Tian et al., 2018a)".

(4) In line 75, "is" have corrected into "are".

(5) In line 121, "desorption energies ($\Delta E_d$)" have corrected into "($\Delta E_d$)"。

(6) In line 145, "this" have corrected into "This".

(7) In line 158, "which" have corrected into "which is".

**Question** 2. There are some inconsistent statements in the paper: "radiative forcing effect" in line 1 and "radiative forced effect" in line 12; "low oxygen-content" in line 164, "low oxygen content" in line 175, and "high-oxygen-content" in line 251; "amine-Kao particles" in line 192 and "amine-Kao particle" in line 194. "Kao surface" in line 86 and "the Kao surface" in line 88; "in gas phase" in line 118, "in the gas phase" in line 123, and "in gaseous phase" in line 190. Please check other sentences and ensure the consistent statement for the same content.

**Response:** We have unified the consistent statement for the same content.

(1) In line 12, "radiative forced effect" has corrected into "radiative forcing effect".

(2) In line 174, "Low oxygen-content" has corrected into "Low-oxygen-content".

(3) In line 222, "amine-Kao particle" has corrected into "amine-Kao particles".

(4) In line 88, "the Kao surface" have corrected into "Kao surface".

(5) There are "in the gas phase" in line129 and "in gaseous phase" in line 218 have corrected into "in gas phase".

**Question** 3. Lines 125-126: The author states "…and the concentration sharply increases when Z is smaller." Is it smaller than 4 Å? Please add the description for clarity.

**Response:** In line 131-132, we have clearly restated the sentence. It was corrected into "…and the concentration sharply increases when Z is smaller than 4.0 Å.".

**Question** 4. Line 146: "…because its hydrogen is completely substituted by methyl group" is suggested to be revised into "…because its hydrogen in amido group is completely substituted by methyl group", which is more accurate.

**Response:** In line 155-156, the sentence "…because its hydrogen is completely substituted by methyl group" have revised into "…because its hydrogen in amido group is completely substituted by methyl group".

**Question** 5. In line 155, The sentence that "the related $k$ for heterogenous initial oxidation reaction of DMA is calculated as 9.22 $s^{-1}molec^{-1}cm^3$, which is 11 orders smaller than that of the gaseous one" is confusing. According to the above sentence, the k of the heterogenous reaction should be larger than that of the gaseous one. They should carefully check it.

**Response:** In line 164, the sentence "smaller than that of the gaseous one" have revised into "higher than that of the gaseous one".

**Question** 6. Lines 165-166: I cannot understand the meaning "…, the OS of each RNCHO increases by 4…". How to reflect the increase of the OS? Please add some discussions.

**Response:** The meaning of OS is defined in lines 108-112. "OS = $n_{C-O}$ - $n_{C-H}$ - $n_{N-H}$, where $n_{C-O}$, $n_{C-H}$ and $n_{N-H}$ indicate the number of C-O, C-H and N-H bonds of amine and its oxidized products". The OS increased by 4 under the pollution condition, referring to the oxidation process of amines to amides, the OS value increased from -5 to -1 (MA), -7 to -3 (DMA), and -9 to -5 (TMA).

**Question** 7. In Figure 6, based on their discussion, the caption should be revised into "Related profiles are for (A, B) MA-Kao (C, D) DMA-Kao and (E, F) TMA-Kao mixed particles, respectively."

**Response:** We've repositioned the graphics and the title of Figure 6 was corrected as "Related profiles are for (A, D) MA-Kao (B, E) DMA-Kao and (C, F) TMA-Kao mixed particles, respectively.".

**Question** 8. In lines 173-174, the authors stated that MA-RO$_2$ cannot be oxidized by O$_2$ under clean condition, but instead it is oxidized by NO even though low concentration NO. The authors should provide specific NO concentration according to their calculated kinetics.

**Response:** According to reference (Berndt et al., 2022), the intermediate of RO$_2$ can undergo addition reaction with ·OH to form more stable molecular products. In lines 185-189, we have added the reaction mechanism of MA/DMA-RO$_2$ under clean conditions. The high oxygen derivatives (R-NCH2OOOH) were produced by the autoxidation and ·OH addition reaction under clean conditions. And data and discussions on OS (Fig. 4), optical properties (Fig. 5-7) and radiative forcing effects (Fig. S6-S7) of MA/DMA-RO$_2$ under clean condition are also supplemented. The final radiative forcing effect results are consistent with the original.

NO concentration is smaller than $1.5 \times 10^{-12}$ molec·cm$^{-3}$ under clean condition and higher than 7.2 molec·cm$^{-3}$ under polluted condition. The two NO concentrations are determined by our thermodynamics and kinetics data (Line 209-211), which are consistent with atmospheric NO concentrations obtained from (Tan et al., 2018). The detailed discussions of NO concentration calculations are provided in Line 199-214.

---

## Author Response (AR2)

**Public justification:**

Line 124, it was stated that "the concentration of Kao is $6.1 \times 10^{21}$ molec·cm$^{-3}$". The unit should be "cm$^{-3}$", instead of "molec·cm$^{-3}$".

**Response:** Thanks for your comment. In Line 126, the unit is corrected into "cm$^{-3}$", which is marked in red in the revised manuscript.